# Examining influential factors in newly diagnosed cancer patients and survivors: Emphasizing distress, self-care ability, peer support, health perception, daily life activity, and the role of time since diagnosis

**Yeganeh Shahsavar*[◐], Avishek Choudhury[ID]\*[◐]**

Industrial and Management Systems Engineering, Benjamin M. Statler College of Engineering and Mineral Resources, West Virginia University, Morgantown, West Virginia, United States of America

◐ These authors contributed equally to this work.
* ys00022@mix.wvu.edu (YS); avishek.choudhury@mail.wvu.edu (AC)

**Data Availability Statement:** The data underlying the results presented in the study are available within the manuscript and its Supporting

## Abstract

This study investigates the complex interrelationships between peer support, mental distress, self-care abilities, health perceptions, and daily life activities among cancer patients and survivors while considering the evolving nature of these experiences over time. A cross-sectional survey design is employed, utilizing de-identified data from the National Cancer Institute's 2022 nationally representative dataset, which comprises responses from 1234 participants, including 134 newly diagnosed patients undergoing cancer treatment. Partial least squares structural equation modeling is employed for data analysis. The results reveal that peer support significantly reduces mental distress and positively influences the perception of self-care abilities and health perceptions among cancer patients and survivors. Additionally, the study finds that mental distress negatively affects daily life activities and self-care abilities. This means that when cancer patients and survivors experience high levels of mental distress, they may struggle with everyday tasks and find it challenging to care for themselves effectively. The research also shows that mental distress tends to decrease as time passes since diagnosis and health perceptions improve, highlighting the resilience of cancer patients and survivors over time. Furthermore, the study uncovers significant moderating effects of age, education, and income on the relationships between daily life activity difficulties, perception of self-care ability, and perception of health. In conclusion, this research provides a comprehensive understanding of the intricate associations between the variables of interest among cancer patients and survivors. The findings underscore the importance of peer support and targeted interventions for promoting well-being, resilience, and quality of life in this population, offering valuable insights for healthcare providers, researchers, and policymakers. Identifying moderating effects further emphasizes the need to consider individual differences when designing and implementing support systems and interventions tailored to the unique needs of cancer patients and survivors.

Information files. The author cannot grant requests for any other underlying data because the data were acquired from the Health Information National Trends Survey (HINTS) under the Microdata Dissemination Policy. Individuals and organizations wishing to access the data can make a request directly to the HINTS service- https://hints.cancer.gov/data/restricted-data.aspx

**Funding:** The author(s) received no specific funding for this work.

**Competing interests:** The authors have declared that no competing interests exist.

## Introduction

Cancer is a complex and heterogeneous disease that varies in severity and prognosis. The severity of cancer and a patient's prognosis depends on several factors, including the type and stage of cancer, the patient's age and overall health, and the effectiveness of treatment. Some types of cancer are more aggressive and have a poorer prognosis, while others may be more slow-growing and have a better prognosis. Despite being a potentially fatal disease, advances in cancer treatment have significantly improved survival rates for many types of cancer. Because of the improving medical sciences, in 2022, 69% of cancer survivors have lived more than five years since their diagnosis, 47% have lived more than ten years, and 18% have lived at least 20 years since their diagnosis [1]. Besides, the number of cancer survivors in the United States is projected to increase by 24.4%, to 22.5 million, by 2032 [1].

Despite improving survival rates, cancer incidence continues to rise [2]. In 2018, the United States had approximately 623,000 people diagnosed with metastatic breast, prostate, lung, colorectal, bladder cancer, or metastatic melanoma. This number is projected to increase to 693,452 by 2025 [3]. Given the aging population, it is estimated that the annual number of cancer cases will increase by 49%, from 1,534,500 in 2015 to 2,286,300 in 2050, primarily affecting adults aged 75 years and older [4]. Such incidences are one of many factors shaping cancer perception and inducing cancer stigma. Cancer stigma refers to the negative attitudes, behaviors, and beliefs directed toward individuals with cancer or survivors [5–7]. Cancer stigma can take many forms, including social stigma, which refers to the negative attitudes and beliefs that society holds about cancer and those who have it, and self-stigma, which refers to the negative attitudes and beliefs that individuals with cancer may hold about themselves. It can lead to feelings of shame, isolation, and discrimination, hindering a person's quality of life and ability to cope with the challenges of cancer and its treatment.

Besides the stigma, the experience of living with cancer is challenging in many ways. Its treatment can significantly impact a person's physical, emotional, and psychological well-being. Cancer patients may experience various physical and psychological challenges, such as fatigue, pain, loss of appetite, fear, anxiety, and depression. These challenges also impact cancer survivors. Survivors report feeling grateful for the opportunity to continue living, but their worries about the possibility of cancer returning and the hurdles in adjusting to life after cancer treatment are substantial. The literature acknowledges that cancer patients and survivors are prone to mental distress, which can deter their quality of life [8–11]. According to a multi-center study, lung cancer survivors who experience poor quality of life also face mental health problems, hurdles in performing everyday physical tasks, and experienced social isolation [12]. Evidence also shows that cancer hinders daily self-care, domestic work, and leisure activities [13–15]. Due to negative psychological impact and hindered quality of life, cancer patients and survivors have reported poorer health perception [16, 17].

Not much study has been done to understand the complex factors that impact the life and perception of cancer patients and survivors. The novelty of this work lies in its comprehensive and integrative approach to understanding the interplay between ease of daily life activity, mental distress, peer support, struggle with self-care, and time since diagnosis. While previous studies have individually examined the impact of peer support, mental distress, and self-care on cancer patients' experiences and outcomes, this research builds upon the existing literature by investigating these factors concurrently and holistically [18]. Our study provides a more nuanced understanding of the intricate associations between these variables and their effects on patients' overall well-being. Furthermore, this work contributes to the growing body of research on the temporal aspects of cancer care and survivorship, considering the evolving nature of patients' experiences over time. By examining the dynamics of mental distress and

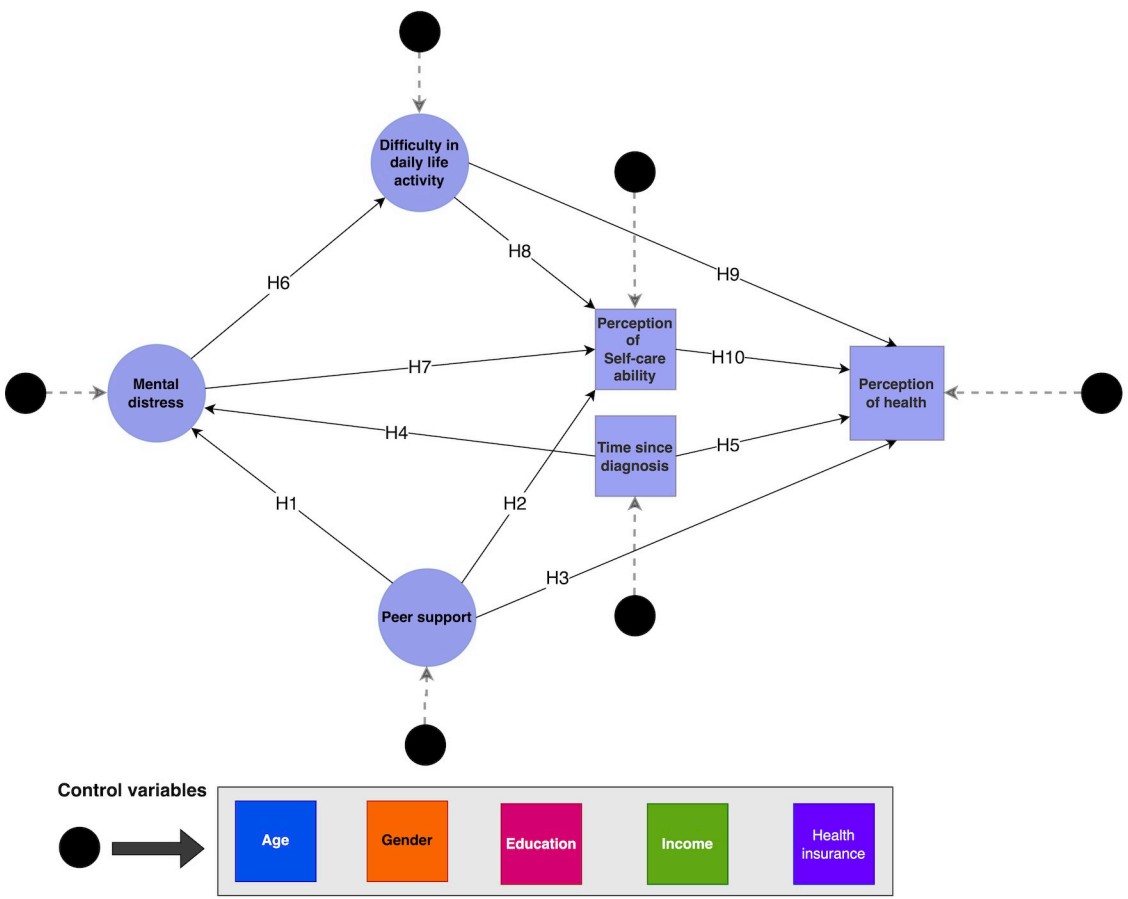

**Fig 1. Conceptual framework illustrating the relationships between 'time since diagnosis,' 'mental distress,' 'difficulty in daily life activity,' 'peer support,' 'perception of self-care ability,' and 'perception of health'.**

health perceptions as time progresses, this study sheds light on the long-term implications of cancer diagnosis and treatment, offering valuable insights for healthcare providers and policy-makers in tailoring long-term care plans and support systems.

This study investigates the following hypotheses, as illustrated in Fig 1:

- Hypothesis 1 (H1): Patients with more peer support will have lower mental distress.

- Hypothesis 2 (H2): Patients with peer support will positively perceive self-care abilities.

- Hypothesis 3 (H3): Patients with peer support will positively perceive health.

- Hypothesis 4 (H4): The mental distress in cancer patients and survivors reduces with time.

- Hypothesis 5 (H5): Perception of the health of cancer patients and survivors improves over time.

- Hypothesis 6 (H6): Patients with mental distress are likelier to struggle with daily life activities.

- Hypothesis 7 (H7): Patients with mental distress are likelier to perceive their self-care abilities negatively.

- Hypothesis 8 (H8): Patients who struggle with daily life activity are likelier to perceive their self-care ability negatively.

- Hypothesis 9 (H9): Patients who struggle with daily life activity are likelier to perceive their health negatively.

- Hypothesis 10 (H10): Patients with a negative perception of self-care ability are likelier to have a negative perception of health.

Exploring the proposed hypotheses is of paramount importance for cancer patients and survivors for many reasons. Firstly, investigating the interrelationships (associations) between factors such as peer support, mental distress, self-care abilities, and health perceptions (H1-H3, H6-H10) contributes to a holistic understanding of the cancer experience. This comprehensive perspective enables healthcare professionals and researchers better to address the diverse needs of cancer patients and survivors, ultimately improving their overall well-being.

Secondly, understanding these hypotheses allows healthcare providers to tailor care plans and support systems to the individual needs of cancer patients and survivors. Personalized interventions can be designed and implemented by identifying the factors significantly influencing well-being. This is particularly relevant for patients who benefit from peer support (H1-H3) and those experiencing mental distress (H6, H7).

Cancer patients and survivors face numerous physical, emotional, and psychological challenges. By examining the impact of factors such as peer support and mental distress on daily life activities and self-care abilities (H6-H9), healthcare providers can develop targeted coping strategies to help patients and survivors manage these challenges more effectively. Furthermore, investigating the temporal aspects of cancer care and survivorship, such as the progression of mental distress and health perceptions over time (H4, H5), provides valuable insights into the long-term implications of cancer diagnosis and treatment. These insights can inform the development of long-term care plans and support systems, ensuring that patients and survivors receive the necessary care throughout their cancer journey.

Empowering cancer patients and survivors is another crucial aspect of exploring these hypotheses. A better understanding of their experiences can contribute to reducing the stigma associated with cancer and fostering a more supportive environment. Increased awareness of the factors affecting cancer patients and survivors can improve mental health and overall well-being.

Finally, understanding the factors that significantly influence the well-being of cancer patients and survivors can guide policymakers in prioritizing resources and developing policies that address the unique needs of this population. By allocating resources effectively and creating targeted policies, healthcare systems can be better equipped to support cancer patients and survivors throughout their journey.

## Method

The study was approved by the institutional review board of West Virginia University, Morgantown, West Virginia, United States (Protocol # 2212691613).

### Data source

In this study, we used the 2022 Health Information National Trends Survey (HINTS), a nationally representative de-identified survey of adults in the United States conducted by the National Cancer Institute (NCI), as our data source [19]. The HINTS data used in this study was restricted and obtained with the NCI's approval. The minimal dataset used in this study

has been provided as a supporting document (**S1 Dataset**). HINTS has been administered periodically since 2003 and aims to assess attitudes, behaviors, and knowledge related to cancer and cancer prevention in the United States [20]. The survey includes a wide range of questions covering various aspects of cancer, including risk behaviors, screening behaviors, attitudes toward cancer and cancer prevention, access to cancer-related information, attitudes toward cancer research, and perceived barriers to cancer prevention and early detection. HINTS is a valuable resource for researchers and policymakers interested in understanding trends in cancer-related attitudes, behaviors, and knowledge in the United States. It can inform the development of cancer prevention and control programs and identify areas where additional research is needed [18].

## Survey instruments and latent constructs

We used 13 observed variables from the survey to feed the proposed conceptual model (Fig 1). Four of these questions were combined to form the latent construct, Ease of Daily Life Activity, **(a)** *Because of a physical, mental, or emotional condition, do you have serious difficulty concentrating, remembering, or making decisions*? **(b)** *Do you have serious difficulty walking or climbing stairs*? **(c)** *Do you have difficulty dressing or bathing*? **(d)** *Because of a physical, mental, or emotional condition, do you have difficulties doing errands alone, such as visiting a doctor's office or shopping*? These questions were selected based on their relevance to the Ease of Daily Life Activity, which refers to the degree to which individuals can engage in everyday activities without difficulty. Responses were recorded as "Yes" and "No."

From the remaining 9, 4 from the Patient Health Questionnaire (PHQ-4) were used to access respondents' mental health status. In our study, this calculated variable was termed– Mental Distress. The PHQ-4 is a self-report measure of mental health that consists of two questions assessing depression and two questions evaluating anxiety [21, 22]. Each question asks the respondent to rate the frequency of specific symptoms over the past two weeks, using a four-point Likert scale ranging from "not at all" to "nearly every day." The PHQ-4 has good reliability and validity and is widely used to assess mental health in research and clinical settings [21, 22].

The latent construct, Peer Support, was formed using two questions: **(a)** how often do you feel you have much *in common with the people around you*? **(b)** *How often do you feel close to people*? The responses were scored on a scale ranging from "never" to "always," with higher scores indicating greater levels of Peer Support. These variables were selected based on their relevance to Peer Support, which refers to the sense of belonging and connection that individuals feel with the people around them. All four latent constructs' convergent, reliability, and discriminant validity were validated.

The remaining three observed variables in Fig 1 were measured using single questions. The perception of Struggle with Self-Care was measured using a five-point Likert scale ranging from "completely confident" to "not confident at all": *Overall, how confident are you about your ability to take good care of your health*?

The Perception of Bad Health was measured using a five-point Likert scale ranging from "excellent" to "poor": *In general, would you say your health is*? Finally, Time Since Diagnosis was measured using the question: *How long ago were you diagnosed with cancer*? The response was measured in years.

## Measurement model

All the analyses were conducted using SEMinR package [23] in R [24]. The convergent validity of the latent constructs was determined based on Cronbach's alpha and composite reliability

($\rho_a$ & $\rho_c$) greater than 0.70 and outer loading greater than 0.50 [25]. We examined the discriminant validity per the indicator according to Fornell-Larcker's criterion [26]. As a complementary assessment, we analyzed the Heterotrait-monotrait (HTMT) ratio less than 0.85 [27]. We also checked for multicollinearity using the variance inflation factor (VIF). All missing values were handled using the pairwise deletion method.

### Structural equation modeling

We used the non-parametric Partial Least Squares-Structural Equation Modeling (PLS-SEM) to test our conceptual model. It is a statistical method used to examine the relationships between observable variables and latent constructs in a conceptual framework [28]. It allows researchers to simultaneously estimate multiple and interrelated dependent relationships between variables and latent constructs. The model was controlled for confounding factors such as respondents' annual income, education level, age, health insurance, and race. These variables were used as covariates in the model, as shown in Fig 1.

We used model-fit to measure the R-squared and adjusted R-squared values to indicate the proportion of variance explained by the model [29]. We used the bootstrapping method with 10000 subsamples [30]. This method involves sampling with replacement from the original sample to create a new sample and then to estimate the model on the new sample. This process is repeated multiple times to create a distribution of estimates, which allows for more accurate inferences about the population. Significance was tested at a 95% confidence level (two-tailed).

Lastly, we conducted a multigroup analysis to compare the observed effects between newly diagnosed cancer patients undergoing treatment and survivors.

## Results

### Participants and measurement model

Of all the 1234 respondents, 1188 had some health insurance, and 134 were newly diagnosed cancer patients undergoing treatment. The mean age of all the respondents was 55.5 years. Of all the survivors, 264 were slightly worried about experiencing a recurrence of cancer, 309 expressed some concerns, and about 249 were moderately to extremely concerned with the problem. Almost half of the respondents stated they faced financial problems due to cancer treatment (n = 544).

Patients undergoing cancer treatment reported experiencing side effects of the treatment: cognitive impairments (n = 49), neuropathy (n = 36), severe fatigue (n = 60), or nausea (n = 34). Some (n = 40) also reported experiencing hair loss, joint pain, myalgia, burns on the skin from 5-fluorouracil cream, hot flashes, osteopenia, bowel issues, creaky joints from estrogen suppressant, myoclonic seizure, hand/foot syndrome ascites, severe headaches, loss of sex drive, shortness of oxygen & breath, cardiac dysfunction, and other. Table 1 shows the number of respondents and their socio-demographic information who respondent to the corresponding questions.

Table 2 shows the loadings and reliability of the latent constructs. All the factor loading was above 0.50 and met reliability criteria.

We then assess the discriminant validity of the constructs using the Fornell-Larcker criteria, as shown in Table 4. To complement Fornell-Larcker criteria, we also conducted the HTMT test.

All HTMT ratios were below the recommended threshold of 0.85 for all pairs of constructs, indicating good discriminant validity (Table 3).

**Table 1. Participant characteristics and socio-demographics (n = 1234).**

| | Newly Diagnosed undergoing treatment (n = 134) | Cancer Survivors (n = 1100) |
|---|---|---|
| **Biological sex** | | |
| Male | 62 | 432 |
| Female | 67 | 513 |
| **Age (years)** | | |
| Adults less than 50 | 3 | 36 |
| 50 to 64 | 26 | 198 |
| 65 to 75 | 40 | 333 |
| 75 and older | 58 | 374 |
| **Education** | | |
| Less than high school | 3 | 27 |
| Highschool graduate | 22 | 108 |
| College | 37 | 251 |
| Bachelor's degree | 35 | 262 |
| Post-Baccalaureate degree | 32 | 295 |
| **Employment status** | | |
| Employed full-time | 14 | 190 |
| Employed part-time | 5 | 55 |
| Homemaker | 6 | 34 |
| Retired | 89 | 601 |
| Other | 14 | 56 |
| **Race** | | |
| Non-Hispanic White | 99 | 716 |
| Non-Hispanic Black | 2 | 12 |
| Hispanic | 13 | 102 |
| Other | 8 | 63 |
| **Annual household income** | | |
| Less than $9,999 | 3 | 18 |
| $10,000 to $14,999 | 5 | 33 |
| $15,000 to $19,999 | 5 | 23 |
| $20,000 to 34,999 | 25 | 98 |
| $35,000 to 49,999 | 9 | 111 |
| $50,000 to $74,999 | 25 | 152 |
| $75,000 to $99,000 | 19 | 159 |
| $100K to $199,000 | 32 | 240 |
| $200K and more | 9 | 121 |

Table 4 shows the Variance Inflation Factors (VIF) for the variables included in the structural model. The VIF values were below the threshold of 2.5, indicating that collinearity was not a significant issue in the analysis.

## Structural model

Our models demonstrated varying degrees of fit, with the perception of health model exhibiting the highest adjusted R-squared value (0.412), indicating that the predictor variables explained 41.2% of the variance in this outcome. Conversely, the models for mental distress, peer support, and time since diagnosis displayed a weaker fit with adjusted R-squared values of 0.116, 0.034, and 0.051, respectively.

**Table 2. Factor loadings and reliability.**

| Latent constructs | Loadings | Alpha (>0.70) | rhoC (<0.70) | AVE (>0.50) | rhoA (>0.70) |
|---|---|---|---|---|---|
| Difficulty in daily life activity | 0.634 | 0.711 | 0.822 | 0.538 | 0.714 |
| | 0.754 | | | | |
| | 0.747 | | | | |
| | 0.789 | | | | |
| Mental distress | 0.855 | 0.882 | 0.918 | 0.737 | 0.894 |
| | 0.900 | | | | |
| | 0.838 | | | | |
| | 0.840 | | | | |
| Peer support | 0.899 | 0.722 | 0.877 | 0.782 | 0.729 |
| | 0.896 | | | | |

Table 5 showcases the structural direct paths analysis outcomes, offering empirical evidence for 9 out of 10 proposed hypotheses concerning the relationships among peer support, mental distress, perception of self-care ability, perception of health, time since diagnosis, and difficulty in daily life activities in cancer patients and survivors. The results of our statistical analysis are as follows:

Hypothesis 1 (H1) proposed that patients with more peer support would experience lower mental distress. The analysis supported this hypothesis, revealing a significant negative association between peer support and mental distress ($\beta$ = -0.318, 95% CI [-0.379, -0.256]). Hypothesis 2 (H2) suggested that patients with peer support would perceive their self-care abilities more positively, and our findings corroborated this assertion ($\beta$ = 0.222, 95% CI [0.166, 0.281]). However, Hypothesis 3 (H3) postulated that patients with peer support would perceive their health more positively, did not yield significant results.

Hypothesis 4 (H4) posited that mental distress in cancer patients and survivors would decrease with time. Our analysis supported this hypothesis, demonstrating a significant negative relationship between time since diagnosis and mental distress ($\beta$ = -0.058, 95% CI [-0.114, -0.002]). Hypothesis 5 (H5) proposed that the perception of health in cancer patients and survivors would improve over time, and our findings confirmed this positive relationship ($\beta$ = 0.056, 95% CI [0.009, 0.101]). Hypothesis 6 (H6) suggested that patients with mental distress would be more likely to struggle with daily life activities, and our analysis provided evidence for this association ($\beta$ = 0.316, 95% CI [0.246, 0.387]). Moreover, Hypothesis 7 (H7) postulated that patients with mental distress would be more likely to perceive their self-care abilities negatively, which was supported by our findings ($\beta$ = -0.133, 95% CI [-0.193, -0.074]).

Hypothesis 8 (H8) posited that patients who struggle with daily life activities would be more likely to perceive their self-care abilities negatively. Our findings supported this hypothesis, revealing a significant negative relationship between difficulty in daily life activities and perception of self-care ability ($\beta$ = -0.262, 95% CI [-0.331, -0.192]). Hypothesis 9 (H9)

**Table 3. Discriminant validity using Fornell-Larcker criteria.**

| | Peer support | Mental distress | Difficulty in daily life activity |
|---|---|---|---|
| **Peer support** | 0.884 | na | na |
| **Mental distress** | -0.323 | 0.859 | na |
| **Difficulty in daily life activity** | -0.142 | 0.330 | 0.734 |

na = not applicable

**Table 4. Variance Inflation Factors for multicollinearity test.**

| | PR | TDx | MD | DDLA | PSC | Age | Education | Income |
|---|---|---|---|---|---|---|---|---|
| **Mental distress (MD)** | 1.042 | 1.059 | - | - | - | 1.130 | 1.200 | 1.246 |
| **Perception of Self-care ability (PSC)** | 1.154 | - | 1.250 | 1.227 | - | 1.111 | 1.200 | 1.300 |
| **Perception of health** | 1.142 | 1.061 | - | 1.230 | 1.246 | 1.166 | 1.207 | 1.301 |
| **Difficulty in daily life activity (DDLA)** | 1.022 | - | - | - | - | 1.066 | 1.199 | 1.241 |
| **Peer support (PR)** | - | - | - | - | - | 1.061 | 1.198 | 1.231 |
| **Time since diagnosis (TDx)** | - | - | - | - | - | 1.061 | 1.198 | 1.231 |

predicted that patients who struggle with daily life activities would be more likely to perceive their health negatively, and our analysis confirmed this association (β = -0.241, 95% CI [-0.297, -0.185]). Finally, Hypothesis 10 (H10) suggested that patients with a negative perception of self-care ability would be more likely to have a negative perception of health, which was also supported by our findings (β = 0.450, 95% CI [0.399, 0.499]).

In our analysis, we also considered several control variables, namely age, gender, education, health insurance, and income. The results pertaining to these control variables are presented in Table 5.

The analysis yielded the following noteworthy findings regarding the control variables:

- Age was found to have significant associations with peer support, time since diagnosis, difficulty in daily life activities, self-care, and health perception.

- Gender emerged as significantly related to peer support, time since diagnosis, mental distress, and health perception.

- Education exhibited significant associations with self-care and health perception.

- Health insurance demonstrated significant relationships with self-care and health perception.

- Income revealed significant associations with peer support, difficulty in daily life activities, and health perception.

As shown in Table 6, the indirect effects examined in our analysis can be matched to the hypotheses, providing further insights into the complex relationships among the study variables. By doing so, we can complement and extend our understanding of the direct effects observed in the hypotheses. Indirect effects refer to the influence of a predictor variable on an outcome variable through the intermediary influence of one or more other variables [31].

H1 (Peer support → Mental distress): The indirect effect of peer support on the difficulty of daily life activity through mental distress further supports H1 and illustrates that decreased mental distress due to peer support also contributes to a reduced difficulty in daily life activities.

H2 (Peer support → Perception of self-care ability): The indirect effects of peer support on self-care through mental distress and the difficulty of daily life activity both reinforce H2, demonstrating that peer support's positive impact on self-care is not only direct but also operates through its influence on mental distress and the difficulty of daily life activity.

H3 (Peer support → Perception of health): Although the direct effect was not significant, the indirect effects of peer support on the perception of health through mental distress, self-care abilities, and the difficulty of daily life activity suggest that peer support might still contribute to a more positive perception of health.

**Table 5. Direct effects.**

| Null hypotheses | Path | β | SD | T | CI [5%, 95%] |
|---|---|---|---|---|---|
| H1: Fail to reject | Peer support → Mental distress | -0.318 | 0.031 | -10.131 | [-0.379, -0.256] * |
| H2: Fail to reject | Peer support → Self-care | 0.222 | 0.029 | 7.583 | [0.166, 0.281] * |
| H3: Reject | Peer support → Perception of health | 0.031 | 0.024 | 1.286 | [-0.016, 0.077] |
| H4: Fail to reject | Time since diagnosis → Mental distress | -0.058 | 0.029 | -2.011 | [-0.114, -0.002] * |
| H5: Fail to reject | Time since diagnosis → Perception of health | 0.056 | 0.023 | 2.395 | [0.009, 0.101] * |
| H6: Fail to reject | Mental distress → The difficulty of daily life activity | 0.316 | 0.036 | 8.856 | [0.246, 0.387] * |
| H7: Fail to reject | Mental distress → Self-care | -0.133 | 0.030 | -4.404 | [-0.193, -0.074] * |
| H8: Fail to reject | The difficulty of daily life activity → Self-care | -0.262 | 0.035 | -7.414 | [-0.331, -0.192] * |
| H9: Fail to reject | The difficulty of daily life activity → Perception of health | -0.241 | 0.029 | -8.319 | [-0.297, -0.185] * |
| H10: Fail to reject | Self-care → Perception of health | 0.450 | 0.026 | 17.570 | [0.399, 0.499] * |
| **Control variables** | | β | SD | T | CI [5%, 95%] |
| Age → Peer support | | 0.116 | 0.028 | 4.087 | [0.062, 0.173] * |
| Age → Time since diagnosis | | 0.225 | 0.028 | 7.954 | [0.171, 0.281] * |
| Age → Mental distress | | -0.021 | 0.032 | -0.645 | [-0.085, 0.041] |
| Age → The difficulty of daily life activity | | 0.166 | 0.028 | 5.919 | [0.109, 0.219] * |
| Age → Self-care | | -0.072 | 0.027 | -2.652 | [-0.125, -0.019] * |
| Age → Perception of health | | 0.049 | 0.023 | 2.156 | [0.005, 0.094] * |
| Gender → Peer support | | 0.080 | 0.028 | 2.890 | [0.024, 0.133] * |
| Gender → Time since diagnosis | | 0.108 | 0.029 | 3.744 | [0.052, 0.165] * |
| Gender → Mental distress | | 0.092 | 0.028 | 3.292 | [0.037, 0.146] * |
| Gender → The difficulty of daily life activity | | 0.035 | 0.026 | 1.322 | [-0.017, 0.086] |
| Gender → Self-care | | -0.005 | 0.026 | -0.178 | [-0.056, 0.045] |
| Gender → Perception of health | | -0.054 | 0.023 | -2.365 | [-0.098, -0.008] * |
| Education → Peer support | | 0.009 | 0.031 | 0.280 | [-0.053, 0.070] |
| Education → Time since diagnosis | | 0.046 | 0.031 | 1.489 | [-0.016, 0.106] |
| Education → Mental distress | | -0.027 | 0.032 | -0.846 | [-0.091, 0.035] |
| Education → The difficulty of daily life activity | | -0.031 | 0.031 | -1.004 | [-0.090, 0.030] |
| Education → Self-care | | 0.062 | 0.029 | 2.166 | [0.005, 0.118] * |
| Education → Perception of health | | 0.082 | 0.024 | 3.452 | [0.035, 0.129] * |
| Health insurance → Peer support | | -0.001 | 0.036 | -0.022 | [-0.074, 0.066] |
| Health insurance → Time since diagnosis | | 0.031 | 0.021 | 1.502 | [-0.011, 0.071] |
| Health insurance → Mental distress | | 0.001 | 0.033 | 0.040 | [-0.057, 0.071] |
| Health insurance → The difficulty of daily life activity | | 0.022 | 0.027 | 0.793 | [-0.022, 0.084] |
| Health insurance → Self-care | | 0.043 | 0.018 | 2.395 | [0.007, 0.077] * |
| Health insurance → Perception of health | | 0.070 | 0.026 | 2.656 | [0.019, 0.122] * |
| Income → Peer support | | 0.165 | 0.033 | 5.084 | [0.103, 0.228] * |
| Income → Time since diagnosis | | 0.031 | 0.031 | 0.985 | [-0.030, 0.093] |
| Income → Mental distress | | -0.044 | 0.035 | -1.271 | [-0.112, 0.023] |
| Income → The difficulty of daily life activity | | -0.181 | 0.034 | -5.239 | [-0.247, -0.112] * |
| Income → Self-care | | -0.012 | 0.031 | -0.384 | [-0.072, 0.050] |
| Income → Perception of health | | 0.098 | 0.026 | 3.698 | [0.047, 0.150] * |

The asterisks (*) in the table denote the significant relationships; β = Standardized path coefficient.

SD = Standard deviation

H4 (Time since diagnosis → Mental distress) and H5 (Time since diagnosis → Perception of health): The significant indirect effects of time since diagnosis on the perception of health through mental distress, the difficulty of daily life activity, and self-care support both H4 and

**Table 6. Indirect effects.**

| Path coefficients | β | SD | T | CI [5%, 95%] |
|---|---|---|---|---|
| Indirect effects of study variables | | | | |
| Peer support → Mental distress→ The difficulty of daily life activity | -0.100 | 0.016 | -6.242 | [-0.134, -0.071] * |
| Peer support →Mental distress→ The difficulty of daily life activity →Self-care | 0.026 | 0.005 | 4.972 | [0.017, 0.037] * |
| Peer support→Mental distress→ The difficulty of daily life activity →Self-care →Perception of health | 0.012 | 0.003 | 4.710 | [0.008, 0.017] * |
| Peer support→Self-care →Perception of health | 0.100 | 0.015 | 6.875 | [0.073, 0.130] * |
| Time since diagnosis →Mental distress→ The difficulty of daily life activity →Self-care →Perception of health | 0.002 | 0.001 | 1.799 | [0.000, 0.005] * |
| Time since diagnosis →Mental distress→The difficulty of daily life activity →Self-care | 0.005 | 0.003 | 1.811 | [0.000, 0.010] * |
| Mental distress→ The difficulty of daily life activity →Self-care | -0.083 | 0.015 | -5.681 | [-0.113, -0.056] * |
| Mental distress→ The difficulty of daily life activity →Self-care→Perception of health | -0.037 | 0.007 | -5.296 | [-0.052, -0.024] * |
| Mental distress →Self-care →Perception of health | -0.060 | 0.014 | -4.214 | [-0.089, -0.032] * |
| Mental distress → The difficulty of daily life activity →Perception of health | -0.076 | 0.014 | -5.451 | [-0.106, -0.051] * |
| The difficulty of daily life activity →Self-care →Perception of health | -0.118 | 0.019 | -6.466 | [-0.155, -0.083] * |

The asterisks (*) in the table denote the significant relationships between the constructs.

β = Standardized path coefficient

SD = Standard deviation

Note: The supporting information, including indirect and total effects of control variables, has been provided as supplementary material accompanying this manuscript (**S1 Dataset**). This supplementary file contains additional analyses and findings that further elucidate the complex relationships among the study variables, offering a more comprehensive understanding of the research topic.

H5, indicating that the positive relationship between time since diagnosis and perception of health also operates through reduced mental distress and improved self-care abilities.

H6 (Mental distress → Difficulty of daily life activity): The indirect effect of mental distress on self-care through the difficulty of daily life activity complements H6, suggesting that the negative impact of mental distress on daily life activities also impairs self-care abilities.

H7 (Mental distress → Perception of self-care ability): The significant indirect effects of mental distress on self-care abilities through the difficulty of daily life activity and the perception of health provide additional evidence for H7, highlighting that mental distress negatively influences self-care abilities not only directly but also through the difficulty of daily life activity and the perception of health.

H8 (Difficulty of daily life activity → Perception of self-care ability) and H9 (Difficulty of daily life activity → Perception of health): The significant indirect effect of the difficulty of daily life activity on the perception of health through self-care reinforces both H8 and H9, illustrating that the negative influence of daily life activity difficulties on self-care abilities and health perception are also interconnected.

These indirect effects provide a more comprehensive understanding of how the study variables interact and complement the direct effects observed in the ten hypotheses. By considering both direct and indirect effects, we can better appreciate the complexity and intricacy of the relationships among peer support, mental distress, perception of self-care ability, perception of health, time since diagnosis, and difficulty in daily life activities in cancer patients and survivors.

## Moderation effect

The significant findings of control variables in our analyses, specifically their influence on the primary variables and outcomes, encouraged us to investigate potential moderation effects. By examining moderation effects, we aim to provide a more comprehensive understanding of the

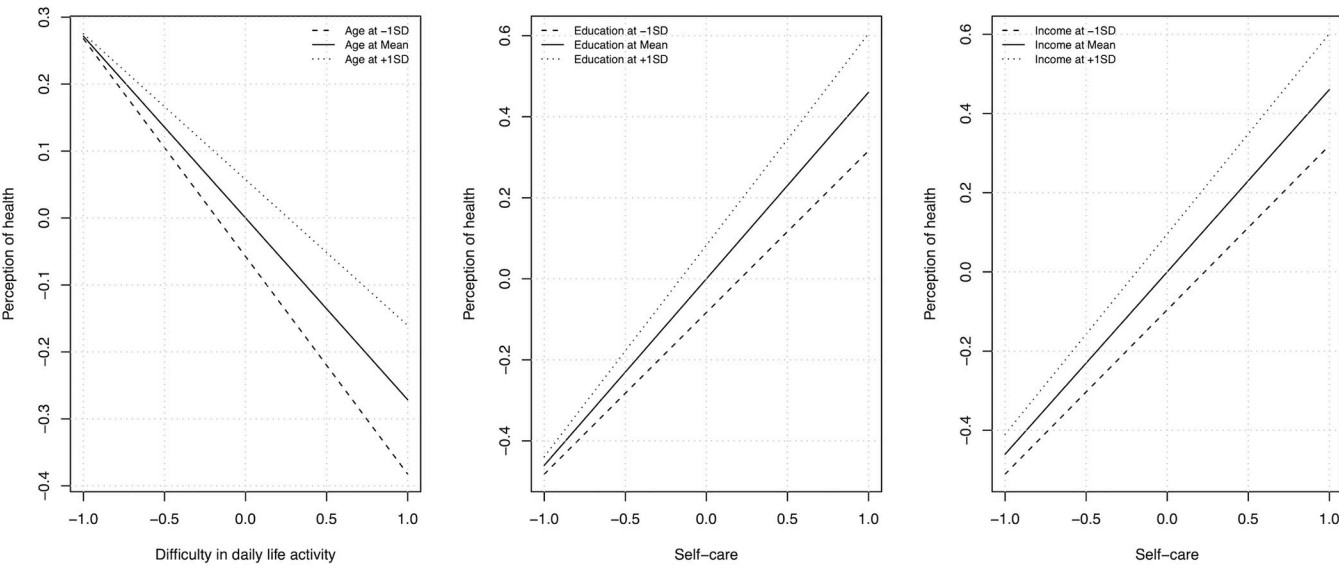

**Fig 2. Moderation effect of age, education, and income.**

relationships among the study variables, considering the influence of control variables on these relationships. In this study, we conducted a simple slope analysis to examine the moderating effects of age, gender, education, income, and health insurance on the relationship between predictor variables (peer support, time since diagnosis, the difficulty of daily life activity, and self-care) and the outcome variable (perception of health) (**S1 File**). Fig 2 illustrates the significant findings of the slope analysis. Table 7 presents the results of the significant moderation effects. The moderation effects identified in our analysis provide valuable insights into how specific factors may influence the relationships among the study variables, thereby adding further depth and nuance to our understanding of the ten hypotheses.

Age moderating the effect of difficulty in daily life activity → Perception of health: This significant moderation effect indicates that the negative relationship between the difficulty of daily life activity and perception of health varies depending on the individual's age. As a result, the impact of daily life activity difficulties on health perception may be more pronounced or mitigated in different age groups. For instance, older patients may find it harder to cope with daily life activities due to declining physical abilities or other age-related factors, resulting in a more negative health perception than younger patients.

Education moderating the effect of perception of self-care ability → Perception of health: The significant moderation effect of education on the relationship between self-care and perception of health suggests that an individual's level of education may influence the extent to which self-care abilities impact their health perception. This finding highlights the need to

**Table 7. The moderating effect of age, education, and income.**

|  | β | SD | T | CI [5%, 95%] |
|---|---|---|---|---|
| Difficulty in daily life activity*Age→ Perception of health | 0.057 | 0.024 | 2.339 | [0.007, 0.102] * |
| Perception of Self-care ability* Education→ Perception of health | 0.069 | 0.024 | 2.807 | [0.021, 0.117] * |
| Perception of Self-care ability * Income→ Perception of health | 0.052 | 0.026 | 2.012 | [0.001, 0.103] * |

The asterisks (*) in the table denote the significant relationships between the constructs.

consider education when evaluating the role of self-care in shaping health perception, as it may strengthen or weaken the relationship depending on the individual's educational background.

Income moderating the effect of perception of self-care ability → Perception of health: The significant moderation effect of income on the relationship between self-care and perception of health demonstrates that an individual's income level can affect how self-care influences their health perception. This implies that the positive association between self-care and health perception may vary across income levels. The relationship may be stronger or weaker depending on an individual's financial situation.

## Multigroup analysis

We conducted a multigroup analysis comparing cancer patients and survivors and found no significant difference between the groups (cancer survivors vs. patients undergoing cancer treatment). The analysis yielded p-values ranging from 0.05 to 0.89 (**reported in S2 File**), indicating no statistically significant differences in the relationships between the two groups' daily life activity, peer support, health perception, and mental distress. This finding implies that the factors influencing the well-being of cancer patients are similar regardless of their cancer status, highlighting the importance of addressing these factors in both patient groups. This result also suggests that survivorship care should not be fundamentally different from cancer care and that interventions designed to improve the quality of life of cancer patients could also benefit survivors. These findings underscore the need for tailored interventions that account for the unique needs and challenges of cancer patients and survivors alike.

## Discussion

### Contribution

The present study offers a unique and comprehensive understanding of the interrelationships between various factors influencing the well-being of cancer patients and survivors. By accounting for the evolving nature of these experiences over time, our study contributes to the existing body of literature on cancer care and survivorship, providing valuable insights into the distinct needs of this population. This novel approach enables the development of tailored interventions and support systems that address the complex needs of cancer patients and survivors, ultimately improving their overall well-being.

The study's findings underscore the significance of individual differences in cancer care, highlighting the importance of interdisciplinary collaboration among healthcare providers to offer comprehensive care that addresses the myriad challenges cancer patients and survivors face. Our results also demonstrate the significant role of financial resources, healthcare access, and social support in shaping cancer patients' well-being, providing policymakers with valuable insights into the factors contributing to disparities in cancer care. By recognizing the importance of these factors, policymakers can design and implement targeted interventions to promote equitable access to healthcare resources and services.

In addition, our study's results highlight the potential benefits of considering the duration since diagnosis when examining the mental health consequences of various health conditions. This novel approach to understanding the effects of cancer on mental health emphasizes the importance of taking into account the unique needs of cancer patients at different stages of their cancer journey. By recognizing the significance of time since diagnosis, healthcare providers can design and implement targeted interventions that improve patients' overall well-being and quality of life, ultimately leading to better outcomes and improved survivorship.

## Main findings

One key finding of our study is the significant association between peer support and reduced mental distress and perception of self-care ability among cancer patients and survivors. Prior evidence also supports our findings, where a study involving patients with prostate cancer reported that peer support fostered resilience, helped individuals regain control, maintained a sense of self, and promoting social connectedness [32]. The significant association identified in our study between the perception of self-care ability reduced mental distress and perception of health is also important. Since the concept of self-care was introduced as an element of nursing theory by Orem [33], many studies have acknowledged its importance in playing a major role in patient health. Many factors, including psychological distress and low self-esteem, have been noted to affect self-care practices [34].

Newly diagnosed cancer patients and survivors typically experience psychological distress, and many interventions and protocols are in place to address this concern [35]. However, studies have not found any significant effect of time since diagnosis on cancer patients' mental health [36, 37]. Our study suggests that as the time since diagnosis increases, the level of mental distress experienced by patients tends to decrease, and the amount of health perception increases. These findings justify why most cancer care protocols [38] and other interventions concerning mental health problems in cancer patients focus on the time of diagnosis or during the early stages of cancer treatment. Nonetheless, the influence of cancer on a patient's mental health is a crucial factor that cannot be ignored. In our study, the mental distress of cancer patients affected the difficulty of daily life activities and self-care. It means that mental distress caused by cancer can considerably impact patients' abilities to carry out daily activities and take care of themselves. This statement highlights the importance of considering the psychological effects of cancer on patients. Our study found that the difficulty of daily life activities negatively influences self-care and health perception among cancer patients, thus emphasizing the importance of considering various aspects of daily life when promoting better self-care and overall well-being for cancer patients and survivors. In line with our findings, a study focusing on older adults in assisted living facilities explored a similar relationship between daily life activities, specifically social engagement, and the psychological well-being of participants while also examining the impact of these factors on self-care and health perception [39].

Compliment prior evidence [40], our study underscores the importance of understanding and addressing the unique challenges older cancer patients and survivors face in managing their daily life activities and self-care practices. According to our study, with increasing age, the difficulty of performing daily life activities also escalates, presenting greater challenges for older individuals in carrying out routine tasks. Concurrently, our study observed a negative impact of age on self-care, signifying that older individuals tend to engage in fewer self-care practices than their younger counterparts. Moreover, our study found a positive association between age, peer support, time since diagnosis, and perception of health. This suggests that older cancer patients may have greater access to or may be more inclined to seek peer support, which could benefit their well-being. Understanding the unique needs of older cancer patients and survivors can help healthcare providers develop tailored interventions and support systems that enhance their overall health and quality of life. A study examining the associations between social support, mental health, and resilience in adolescent and young adult cancer survivors provided insights into the potential benefits of social support in enhancing their well-being [41]. The positive relationship between age and perception of health in our study suggests that older patients may have a more optimistic outlook on their health or are more resilient in the face of health challenges. Furthermore, our analysis revealed a significantly greater negative impact of perceived health on difficulties in daily life activities for patients aged 55.5 and older.

In another study, researchers investigated the prevalence rates of anxiety and depression in cancer patients, taking into account cancer type, gender, and age [42]. Their findings indicated that the prevalence of anxiety and depression varied considerably depending on these factors. Certain cancer types were associated with higher rates of psychological distress, while gender and age also played a significant role in shaping patients' emotional well-being after diagnosis. This underscores the importance of considering individual differences and specific cancer types when addressing the mental health needs of cancer patients and supports our findings regarding the relationship between age and mental distress. Recognizing these factors is crucial in developing targeted interventions and support systems to improve cancer patients' and survivors' mental health and overall well-being.

Our study found that gender has a positive relationship with time since diagnosis and a negative association with the perception of health. Additionally, we discovered that gender influences peer support, with females tending to have higher levels of peer support than males. In other words, female cancer patients are more likely to seek and receive peer support, which may benefit their well-being. A previous study revealed gender-specific differences in the quality of life for older cancer patients, with certain aspects of well-being being more affected in one gender compared to the other [43]. This highlights the importance of considering gender differences when designing support systems and interventions for cancer patients, as their needs and experiences may differ based on gender. Tailoring care and support to address these differences can contribute to improved well-being and overall quality of life for cancer patients and survivors.

Our study found a positive association between higher levels of education and self-care and the perception of health. Additionally, we observed that the positive influence of perceived health on self-care was significantly higher among patients with a bachelor's degree or higher educational attainment. Cancer patients with limited literacy skills may face challenges comprehending medical information and communicating effectively with their healthcare providers. This could result in heightened stress levels and difficulties in their daily activities. In line with our findings, a previous study discovered that education and income significantly impact cancer patients' self-care, perception of health, and quality of life [44]. These results emphasize the importance of considering educational background when designing interventions and support systems for cancer patients. Providing accessible information and resources tailored to different levels of education can help empower patients, facilitate better communication with healthcare providers, and ultimately improve their self-care practices, health perception, and overall well-being.

In our study, we observed the considerable impact of medical insurance coverage and income on the overall well-being of cancer patients. We demonstrated that health insurance influences self-care, possibly because it provides access to healthcare resources, information, and services that support and promote self-care activities. Our findings revealed a significant association between health insurance status and income levels, and patients' self-perceptions of their health. Specifically, individuals lacking medical insurance were more predisposed to perceiving their health as poor.

These results emphasize the crucial role financial resources and healthcare access play in shaping cancer patients' well-being, thereby underscoring the need for policy interventions to address these disparities and ensure equitable healthcare access. A 2017 study reported the importance of health insurance in the cancer care [45]. According to their study, health insurance reduced the likelihood of healthcare disparities among cancer patients from underserved communities such as rural Appalachia. Additionally, patients with financial instability may struggle to access and afford healthcare services and experience financial worries, which can impact their mental health. Similarly, an earlier investigation found a strong correlation

between daily life activities and education level, income, and gender [46]. The interplay between these socioeconomic factors and cancer patients' well-being highlights the importance of developing comprehensive, inclusive policies and interventions that cater to the diverse needs of cancer patients across different demographic groups. This could involve increasing access to affordable health insurance, providing financial support for cancer patients, or offering targeted educational programs to improve health literacy and promote self-care practices among vulnerable populations. By addressing these socioeconomic disparities, we can work towards ensuring that all cancer patients have the resources and support they need to manage their condition and maintain their well-being.

Our study also found an association between higher income levels and increased access to peer support. Patients with higher income levels may have better access to peer support due to their ability to join support groups or networks or their capacity to invest in resources that facilitate connections with others who share similar experiences. A previous study found that patients who received peer support were more likely to follow treatment guidelines and engage in multidisciplinary care. Factors associated with increased exposure to peer support included being younger, having a higher income, and being in a committed relationship, such as marriage or a domestic partnership [47]. Moreover, we observed a relationship between lower income levels and increased difficulty in daily life activities. This could be due to financial constraints, limited access to healthcare resources, or the challenges of living in disadvantaged socioeconomic conditions. Our findings emphasize the importance of addressing socioeconomic disparities in cancer care to promote better health outcomes and overall well-being among cancer patients. This may involve implementing policies that improve access to healthcare resources, financial support, and peer support programs for patients with lower income levels. By taking these steps, we can ensure that all cancer patients have the resources and support to manage their condition, engage in self-care practices, and maintain a higher quality of life.

### Future work and limitations

There are several potential areas for future work based on the findings of this study. First, it may be beneficial to conduct additional research to explore further the mechanisms through which demographic and socioeconomic factors influence self-care, mental distress, and overall well-being in cancer patients. This could involve more in-depth analyses of specific factors such as income, education level, and insurance status, as well as investigating the impact of cultural and social norms on these relationships.

Second, future studies could aim to validate and extend the findings of this study by using more rigorous research designs, such as longitudinal studies or randomized controlled trials. These studies could also consider a wider range of outcomes, such as treatment adherence, quality of life, and survival rates, in addition to the factors examined in this study.

Third, exploring the impact of specific interventions on the outcomes studied in this research may be useful. For example, future studies could investigate the efficacy of interventions such as peer support groups, self-care education programs, and financial assistance programs in improving the well-being of cancer patients and survivors. This could involve testing the effectiveness of these interventions across different demographic groups and cancer types.

It is important to acknowledge that this study has some potential limitations that must be considered. One of the limitations is that the data was obtained through self-reported assessments, which can be influenced by biases such as social desirability bias and individual perception. Therefore, the data may not reflect cancer patients' experiences and behaviors accurately. Moreover, our study was conducted with the US population, which may limit the

generalizability of our findings to other settings. Further studies are needed to validate this study's results and explore the impact of specific interventions focusing on the time since diagnosis or breaking poor news on the mental health and well-being of cancer patients. For instance, future studies could examine the effectiveness of cognitive-behavioral or mindfulness-based interventions in improving the mental health outcomes of cancer patients. Additionally, studies can investigate the role of social support groups in enhancing self-care and mental health among cancer patients.

Despite the potential limitations, our study provides valuable insights into the complex interplay between various factors affecting cancer patients' mental health and well-being. These findings can inform the development of interventions and policies that support cancer patients and survivors, promoting better self-care practices, mental health, and overall quality of life.

## Conclusion

Our study findings have significant implications for the overall treatment of individuals with cancer. Prior research has indicated that interventions aimed at reducing social isolation, enhancing self-perception, increasing daily life activities, and reducing psychological distress may positively impact the lives of cancer patients. This study provides new insights and reaffirms the complex interrelationships between daily life activity, peer support, health perception, and cancer patients' mental distress. As such, interventions targeting these factors may help to improve cancer patients' overall care quality.

Our study highlights the importance of prioritizing mental health screenings in the cancer care process, and healthcare providers should also consider the impact of daily life activities on cancer patients. By addressing these concerns, healthcare providers can support cancer patients' overall well-being, enhancing their quality of care. Moreover, our findings underscore the need for interdisciplinary collaboration among healthcare providers, including oncologists, mental health professionals, and other specialists, to offer comprehensive care that addresses the myriad challenges cancer patients and survivors face.

In conclusion, our study offers new insights into the dynamics of cancer care and survivorship, demonstrating the importance of considering individual differences and the impact of demographic and socioeconomic factors on self-care and daily life activities. By providing a more nuanced understanding of the complex interrelationships between various factors influencing the well-being of cancer patients and survivors, our study offers the potential to develop tailored interventions and support systems that address the distinct needs of this population.

## Supporting information

**S1 Dataset. Minimal dataset.**
(CSV)

**S1 File. Extra analysis and results.**
(DOCX)

**S2 File. Multigroup analysis.**
(DOCX)

## Author Contributions

**Conceptualization:** Avishek Choudhury.

**Data curation:** Avishek Choudhury.

**Formal analysis:** Yeganeh Shahsavar, Avishek Choudhury.

**Investigation:** Yeganeh Shahsavar, Avishek Choudhury.

**Methodology:** Yeganeh Shahsavar, Avishek Choudhury.

**Supervision:** Avishek Choudhury.

**Validation:** Avishek Choudhury.

**Visualization:** Yeganeh Shahsavar, Avishek Choudhury.

**Writing – original draft:** Yeganeh Shahsavar, Avishek Choudhury.

**Writing – review & editing:** Yeganeh Shahsavar, Avishek Choudhury.

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
