## [Decision Letter · Decision Letter 0]

26 Jun 2023

PONE-D-23-11115Examining influential factors in newly diagnosed cancer patients and survivors: Emphasizing distress, self-care ability, peer support, health perception, daily life activity, and the role of time since diagnosisPLOS ONE

Dear Dr. Choudhury,

Thank you for submitting your manuscript to PLOS ONE. After careful consideration, we feel that it has merit but does not fully meet PLOS ONE’s publication criteria as it currently stands. Therefore, we invite you to submit a revised version of the manuscript that addresses the points raised during the review process.

We look forward to receiving your revised manuscript.

Kind regards,

Boshra Ismael Ahmed Arnout

Academic Editor

PLOS ONE

Journal Requirements:

Reviewers' comments:

Reviewer's Responses to Questions

**Comments to the Author**

1. Is the manuscript technically sound, and do the data support the conclusions?

Reviewer #1: Yes

2. Has the statistical analysis been performed appropriately and rigorously? 

Reviewer #1: Yes

3. Have the authors made all data underlying the findings in their manuscript fully available?

Reviewer #1: No

4. Is the manuscript presented in an intelligible fashion and written in standard English?

Reviewer #1: Yes

5. Review Comments to the Author

Reviewer #1: this paper reads well and the results are rather interesting.

in line 242 you say that '264 were slightly worried about getting reinfected with cancer'. I don't think cancer is widely recognised as an infectious disease and another word should be used instead of 'reinfected'.

In table 1 the sample size (N) should be added in the title of the table and the 'n' of each group should also be included in the title's row.

6. PLOS authors have the option to publish the peer review history of their article (what does this mean?). If published, this will include your full peer review and any attached files.

Reviewer #1: No

---

## [Author Response · Author response to Decision Letter 0]

26 Jun 2023

Reviewer #1: this paper reads well and the results are rather interesting.

in line 242 you say that '264 were slightly worried about getting reinfected with cancer'. I don't think cancer is widely recognised as an infectious disease and another word should be used instead of 'reinfected'.

Response: We have rephrased the sentence to “264 were slightly worried about experiencing a recurrence of cancer”

In table 1 the sample size (N) should be added in the title of the table and the 'n' of each group should also be included in the title's row.

Response: We have added the sample size to the table caption and group titles in each column.

---

## [Decision Letter · Decision Letter 1]

21 Aug 2023

Examining influential factors in newly diagnosed cancer patients and survivors: Emphasizing distress, self-care ability, peer support, health perception, daily life activity, and the role of time since diagnosis

PONE-D-23-11115R1

Dear Dr. Choudhury,

We’re pleased to inform you that your manuscript has been judged scientifically suitable for publication and will be formally accepted for publication once it meets all outstanding technical requirements.

Kind regards,

Boshra Ismael Ahmed Arnout

Academic Editor

PLOS ONE

Additional Editor Comments (optional):

Reviewers' comments:

Reviewer's Responses to Questions

**Comments to the Author**

1. If the authors have adequately addressed your comments raised in a previous round of review and you feel that this manuscript is now acceptable for publication, you may indicate that here to bypass the “Comments to the Author” section, enter your conflict of interest statement in the “Confidential to Editor” section, and submit your "Accept" recommendation.

Reviewer #1: All comments have been addressed

2. Is the manuscript technically sound, and do the data support the conclusions?

Reviewer #1: (No Response)

3. Has the statistical analysis been performed appropriately and rigorously? 

Reviewer #1: Yes

4. Have the authors made all data underlying the findings in their manuscript fully available?

Reviewer #1: No

5. Is the manuscript presented in an intelligible fashion and written in standard English?

Reviewer #1: Yes

6. Review Comments to the Author

Reviewer #1: (No Response)

7. PLOS authors have the option to publish the peer review history of their article (what does this mean?). If published, this will include your full peer review and any attached files.

Reviewer #1: No

---

## [Editor Report · Acceptance letter]

24 Aug 2023

PONE-D-23-11115R1 

Examining influential factors in newly diagnosed cancer patients and survivors: Emphasizing distress, self-care ability, peer support, health perception, daily life activity, and the role of time since diagnosis 

Dear Dr. Choudhury:

I'm pleased to inform you that your manuscript has been deemed suitable for publication in PLOS ONE. Congratulations! Your manuscript is now with our production department. 

Kind regards, 

on behalf of

Professor Boshra Ismael Ahmed Arnout 

Academic Editor

PLOS ONE